# Exploring a link between the Middle Eocene Climatic Optimum and Neotethys continental arc flare-up

**Annique van der Boon\*[1], Klaudia F. Kuiper[2], Robin van der Ploeg[1a], Margot J. Cramwinckel[1b], Maryam Honarmand[3], Appy Sluijs[1], Wout Krijgsman[1]**

*\*Corresponding author, present address: Geomagnetic Laboratory, Oliver Lodge Building, Department of Physics, Oxford Street, Liverpool, L69 7ZE, United Kingdom, AvanderBoon.work@gmail.com*

*[1] Department of Earth Sciences, Utrecht University, The Netherlands; Princetonlaan 8a, 3584 CB Utrecht, The Netherlands, R.vanderPloeg@uu.nl, M.J.Cramwinckel@uu.nl, A.Sluijs@uu.nl, W.Krijgsman@uu.nl [1a] – now at Shell Global Solutions International B.V., Grasweg 31, 1031 HW Amsterdam, [1b] – now at National Oceanography Centre Southampton, University of Southampton Waterfront Campus, European Way, Southampton, SO14 3ZH, UK*

*[2] Dept. of Earth Sciences, Faculty of Science, Vrije Universiteit Amsterdam, De Boelelaan 1085, 1081 HV Amsterdam, The Netherlands, K.F.Kuiper@vu.nl*

*[3] Department of Earth Sciences, Institute for Advanced Studies in Basic Sciences (IASBS), P.O. Box 45195-1159, Zanjan, Iran, M.Honarmand@iasbs.ac.ir*

**Abstract.** The Middle Eocene Climatic Optimum (MECO), a ~500 kyr episode of global warming that initiated at ~40.5 Ma, is postulated to be driven by a net increase in volcanic carbon input, but a direct source has not been identified. Here we show, based on new and previously published radiometric ages of volcanic rocks, that the interval spanning the MECO corresponds to a massive increase in continental arc volcanism in Iran and Azerbaijan. Ages of Eocene igneous rocks in all volcanic provinces of Iran cluster around 40 Ma, very close to the peak warming phase of the MECO. Based on the spatial extent and volume of the volcanic rocks as well as the carbonaceous lithology in which they are emplaced, we estimate the total amount of $CO_2$ that could have been released at this time corresponds to between 1052 and 12,565 Pg carbon. This is compatible with the estimated carbon release during the MECO. Although the uncertainty in both individual ages, and the spread in the compilation of ages, is larger than the duration of the MECO, a flare-up in Neotethys subduction zone volcanism represents a plausible excess carbon source responsible for MECO warming.

## 1 Introduction

The MECO is characterized by surface and deep ocean warming, both of approximately 2-6°C. MECO warming initiated at ~40.5 Ma, culminating in a short peak warming phase at ~40.0 Ma and terminating at ~39.9 Ma with a comparatively rapid cooling (Bijl et al., 2010; Bohaty et al., 2009; Bohaty and Zachos, 2003; Boscolo Galazzo et al., 2013, 2014; Cramwinckel et

al., 2018). The MECO is associated with a rise in atmospheric $CO_2$ concentrations (Bijl et al., 2010; Henehan et al., 2020), extensive deep sea carbonate dissolution (Bohaty et al., 2009) and marine biotic change (Bijl et al., 2010; Cramwinckel et al., 2019; Edgar et al., 2013; Witkowski et al., 2012). The MECO inherently differs from the early Paleogene transient

warming events such as the Paleocene-Eocene Thermal Maximum (PETM; ~56 Ma) primarily in its longer duration (~500 kyr) of warming, precluding a sudden trigger but rather suggesting a continued driver (Bohaty and Zachos, 2003; Sluijs et al., 2013). Furthermore, unlike the PETM and similar transients, the MECO is not characterized by a negative $\delta^{13}C$ excursion of the exogenic carbon pool, ruling out the input of $^{13}C$-depleted organic-sourced carbon as a driver, but suggesting a volcanic source (Bohaty and Zachos, 2003). Reconstructions and simulations of the carbon cycle indeed point to an

imbalance in the long-term inorganic carbon cycle during the MECO (Sluijs et al., 2013), caused by enhanced volcanism and sustained by diminished continental silicate weathering (van der Ploeg et al., 2018). However, this scenario is quantitatively far from settled, partly because recent analyses based on foraminifer boron isotope ratios suggest that atmospheric $CO_2$ concentrations rose by significantly less than a doubling and did not rise substantially during the onset of the MECO (Henehan et al., 2020). In addition, a plausible source of excess volcanic $CO_2$ remains to be identified.

Here, we explore a volcanic arc flare-up in the Neotethys subduction zone as a potential source. Arc flare-ups can generate 80-90% of the total volume of igneous rocks in arc systems in periods of a few million years (Ducea and Barton, 2007). During the Eocene, a large flare-up took place in vast areas of present-day Iran (see Figure 1A) and these volcanic rocks show subduction-related geochemical signatures, representative of continental arc volcanism (e.g. Moghadam et al., 2015; Pang et al., 2013; Verdel et al., 2011). Geologic settings of the Eocene volcanic regions in Iran differ. Extensive magmatism

in the Lut block is regarded by Pang et al. (2013) to be the result of post-collisional convective removal of the lithosphere and not directly related to subduction. Volcanism in the Sabzevar zone is linked by Moghadam et al. (2016) to lithospheric delamination, possibly assisted by slab-breakoff. In the Talesh/Alborz region, there are conflicting theories on the formation of the volcanic rocks. Asiabanha & Foden (2012) mention a post-collisional transition to a continental arc in their title, but then describe the volcanism as back-arc volcanism. Van der Boon (2017) gives an overview of proposed conflicting settings

for volcanism in the Alborz. It is striking that in most of the areas in Iran, the flare-up is linked to an extensional setting (e.g. Verdel et al., 2011), which makes it different from other flare-ups (e.g. Ducea et al., 2015; Ducea and Barton, 2007).

The main volcanic arc associated with the Neotethys subduction zone stretches from Bazman in southeast Iran towards Azerbaijan in the northwest, from where it continues westwards into Armenia, Georgia and Turkey (Van Der Boon et al., 2017). North of the volcanic arc, in the Peri-Tethys basin of Azerbaijan and Russia, thick bentonites and ash layers are found

within middle Eocene marine sediments (Beniamovski et al., 2003; Seidov and Alizade, 1966). In the past, it has been hypothesised based on field studies, that the middle Eocene part makes up the bulk of the Eocene volcanic succession (e.g. Glaus, 1965), because of the presence of middle Eocene nummulites within the Karaj formation (e.g. Sieber, 1970) and volcanism climaxes during middle Eocene time (Berberian and King, 1981; Davoudzadeh et al., 1997; Verdel, 2009). Sahandi et al. (2014) produced a compilation of geological maps of Iran, which shows that more than half of the outcrop area

of igneous rocks in Iran is of Eocene age (see Figure 1A). The total surface area that is covered by Eocene igneous rocks is

almost 70.000 km$^2$ (including units mapped as Middle Eocene, Eocene-Oligocene, etc.). A causal relationship between peak volcanism in this region and the MECO has been suggested (Allen and Armstrong, 2008; Kargaranbafghi and Neubauer, 2018), but radio-isotopic age constraints to test this hypothesis are insufficient. To quantitatively assess whether volcanism in the Iran-Azerbaijan region could have been a contributor to global warming during the MECO, we present a compilation

of new and previously published radiometric ages for volcanic rocks and estimate eruptive volumes of the flare-up in Iran to evaluate how much $CO_2$ could have been released during this continental arc flare-up.

## 2 Dating the continental arc flare-up of the Neotethys subduction zone

### 2.1 New $^{40}$Ar/$^{39}$Ar data

We analyzed 48 samples of Eocene volcanic rocks of the Azerbaijan-Bazman Arc in Iran and Azerbaijan. Lava flows of the

75 Peshtasar Formation were dated by Vincent et al. (2005) and van der Boon et al. (2017), but ages suffered from severe excess argon. Here, we re-dated lava flows from the lower and middle part of the Peshtasar Formation using new instrumentation to check for potential age bias caused by hydrocarbon interferences in previous data. We further dated samples of two ash layers in the Kura basin in Azerbaijan, as well as four volcanic rocks from the Talesh and western Alborz in Iran (see Figure 1B). Depending on the rock type, groundmass, plagioclase, sanidine, biotite and/or glass was measured

(see Table 1). Thin section analysis showed pervasive alteration of volcanic rocks, disqualifying many sampled units for radio-isotope dating (see supplementary file S1 for a comparison of some thin sections). However, 8 samples showed no significant alteration and were prepared for $^{40}$Ar/$^{39}$Ar dating using standard mineral separation techniques including heavy liquid and magnetic separation and handpicking. In general, fractions between 250-500 μm size were taken. For some minerals, both groundmass or glass and plagioclase or biotite could be separated.

Samples were leached with diluted $HNO_3$ and/or HF. Samples were irradiated during resp. 12 and 18 hours in two irradiations (VU101 in 2014 and VU107 in 2016) at the Oregon State University Triga CLICIT facility, together with Fish Canyon Tuff sanidine as standard (FCs; 28.201 ± 0.023 Ma; Kuiper et al., 2008). After irradiation samples were loaded in Cu-trays and run on a 10-collector Helix-MC mass spectrometer with an in-house built extraction with SEAS NP10, St172 and Ti sponge getters and a Lauda cooler run at -70°C, at the Vrije Universiteit Amsterdam. The used cup-configuration was

either $^{40}$Ar on the H2 Faraday cup and 39-36 argon isotopes on compact discrete dynodes, or both $^{40}$Ar and $^{39}$Ar on respectively H2 and H1 Faraday. Gain calibration was done by peakjumping $CO_2$ in dynamic mode on the different cups (see Monster, 2016 for details). Samples were analyzed using step-heating experiments, while for the ash layers usually single or a few grains were fused in one step and analyzed. Initial measurements were on single or a small number of grains, leading in some samples to very low intensities of $^{40}$Ar (3-4 times higher than blanks). In those cases, more grains were loaded in the

next experiment. Ages are calculated relative to the age of FCs reported in Kuiper et al. (2008; 28.201 ± 0.023 Ma) with decay constants of Min et al. (2000).

Out of the 8 prepared samples, 7 gave results. Our new $^{40}$Ar/$^{39}$Ar ages from igneous rocks and ash layers fall within a range of ~36-45 Ma (Figure 2A), with weighted mean ages per sample between 39.3-43.1 Ma (Figure 2B). Detailed results per sample are described in supplementary file S5, and detailed results per experiment can be found in supplementary files S6-S32. Multiple aliquots of the same samples were measured. The integrated density distribution of these data reveals a peak at around 40.0 Ma (Figure 2B). All compiled ages are shown together with the scaled areal extent of mapped units of Sahandi et al. (2014) (see Figure 2C).

**2.2 Compilation of literature data**

We combined our newly acquired data with more than 420 ages from 72 published studies, including K-Ar, Ar-Ar, U-Pb, Rb-Sr and Re-Os ages (but mainly Ar-Ar and U-Pb; see supplementary files S2 and S3). Our age compilation aimed at pre-Quaternary rocks and is incomplete with respect to Quaternary and pre-Paleogene igneous rocks in Iran. We then used a kernel density plot (Vermeesch, 2012) to integrate all ages from 60-0 Ma, together with our newly acquired data. Ages and their 1σ uncertainties are used as input in the calculation of these distributions. Optimal bandwidth is calculated automatically, and we have set the bin width to 1 Myr. When studies did not report the significance level of their uncertainties, we assumed a 1σ uncertainty. Where possible, Ar-Ar ages were recalibrated to the standard of the Fish Canyon Tuff according to the Kuiper et al. (2008) calibration model. In some cases, original studies did not provide sufficient information for recalibration and then the original ages were used. All details of literature ages and associated references are added in supplementary files S2 and S3.

The compilation of $^{40}$Ar/$^{39}$Ar ages from the literature, mostly from extrusive rocks (only 5 Ar-Ar ages are from intrusive rocks), yields a highly similar age density distribution to our dated samples (see Figure 3A), showing a peak at 39.7 Ma. Published U-Pb ages are typically obtained from zircons which provide less accuracy for eruption ages than $^{40}$Ar/$^{39}$Ar ages from groundmass, plagioclase, sanidine or biotite (Simon et al., 2008), which is reflected in the greater width of the peaks from extrusive U-Pb ages (see Figure 3B). Combined, the Ar-Ar and U-Pb ages obtained from extrusive rocks record a wide peak around 42 Ma, with two sub-peaks at 43.4 and 39.4 Ma. Two smaller peaks at 29.8 and 17.1 Ma are apparent (see Figure 3C). Intrusive activity also peaks around the same time, with radiometric ages from intrusive rocks (n=201) showing a peak at 40.5 Ma, with another sub-peak at 36.6 Ma (Figure 3D). Smaller peaks in intrusive activity are present at 29.7 and 19.9 Ma.

**3 Neotethys volcanism and the MECO**

Considering that the Neotethys subduction zone has been active since the late Triassic (Arvin et al., 2007), our compilation shows a remarkable clustering of ages during the middle Eocene at ~40 Ma. Estimation of the areal extent of middle Eocene volcanic rocks is done using the shapefiles of Sahandi et al. (2014), who made a compilation of geologic maps. According to the geologic maps, 54% of all area covered by volcanic rocks in Iran is of Eocene age. For the Eocene, shapefiles are

classified as 'Eocene', 'Eocene-Oligocene', 'Late Eocene-Oligocene', 'Middle Eocene', and 'Middle-Late Eocene'. More than half is marked as 'Eocene' and not specified further, but of the rest that is specified, almost half is 'Middle Eocene'. Assuming that the unspecified Eocene rocks have approximately the same age distribution as the specified Eocene rocks, we estimate that roughly half of the Eocene volcanic rocks in Iran and a quarter of the total area covered by volcanic rocks in Iran is of middle Eocene age. We use these areas to estimate the volumes of volcanic rocks formed in the middle Eocene. We thus assumed that shapefiles specified as 'Eocene' had the same proportion of middle Eocene igneous rocks, and calculated an areal extent of 38223 km$^2$ of middle Eocene igneous rocks.

Our compilation indicates that many volcanic provinces in Iran were active simultaneously around 40 Ma (see Figure 2C), including the Azerbaijan-Bazman magmatic arc in the west, the Sabzevar zone in northeast Iran (Shafaii Moghadam et al., 2015) and the Lut block in the east (Pang et al., 2013). Some of the largest volumes of middle Eocene volcanic rocks are located in the Talesh Mountains, where 4 out of 5 exposures with the largest areal extent are mapped (marked in white on Figure 1A). Almost three quarters of U-Pb ages ($n_{total}$=329) in Iran are derived from intrusive rocks ($n_{intrusive}$=239). All ages of the intrusive rocks together reveal a peak at ~40.5 Ma (Figure 3D), indicating that the peak of middle Eocene volcanism is also close in time to peak intrusive activity.

It is thus clear that the MECO corresponds to a phase of intense volcanism in the studied area. However, the average error (1σ) of the literature-based ages from 20-60 Ma is 585 kyr, and thus exceeds the duration of the MECO (500 kyr). Furthermore, the exact ages of the peaks in volcanic activity in Figure 2 are sensitive to the number of data points included and are thus not particularly robust – the addition of a few new data points may shift the peaks by thousands of years.

## 4 Volcanic CO$_2$ emissions in Iran and the MECO

The surface area of Iran covered by middle Eocene volcanic rocks is almost 40.000 km$^2$ (Sahandi et al., 2014; Table 2). These volcanic rocks were produced by numerous eruptions throughout the middle Eocene. In the Alborz and Central Iran, middle Eocene volcanic formations are reported to be very thick, with estimates ranging from 3-5 km in the Alborz Mountains (Stöcklin, 1974), to 6-12 km locally throughout nearly all of Iran (Berberian and King, 1981). More recent estimates of the thickness are 3-9 kilometers (e.g. Morley et al., 2009; Verdel et al., 2011). These estimates are supported by geologic maps and their descriptions that are based on extensive fieldwork. Estimates from maps range mostly between 2 and 7 kilometres. On the lower side are for example Saein Qaleh (Kholghi Khasraghi, 1994), Saveh (Ghalamghash et al., 1998a), and Kuhpayeh (Radfar et al., 2002), with thicknesses of ~2 kilometres, Tafresh (Hadjian et al., 1999) with ~3 kilometres, then Meyamey (Amini Chehragh and Ghalamghash, n.d.), Tarom (Hirayama et al., 1966) and Kalateh (Jafarian, n.d.) with around ~4 kilometres, while Kajan (Amini and Amini Chehragh, 2001), Kahak (Ghalamghash et al., 1998b) and Lahrud (Babakhani et al., 1991) mention thicknesses of the Eocene volcanic succession of approximately 6 kilometres, and Bardsir (Mohajjel Kafshdouz and Khodabandeh, 1992) of around 7 kilometres. On the other hand, Iwao and Hushmand-Zadeh, (1971) show a generalised lithostratigraphic log of the Karaj formation, and mention that the succession reaches a thickness of more than 10 kilometres in the Alborz Mountains. In Table 2, we calculate how much CO$_2$ could have been

released through formation of different volumes of volcanic rocks. We calculate this for a range of thicknesses between 2 and 10 kilometres. Extrapolating these thicknesses, this implies a total volume of middle Eocene volcanic rocks between $7.6*10^4$ and $3.8*10^5$ $km^3$ (see Table 2) that potentially produced significant amounts of $CO_2$. Our estimates of $CO_2$ release due to middle Eocene volcanism in Iran are likely underestimates, as there is volcanism in other regions along the Neotethys subduction zone. Unfortunately, the lack of shapefiles of Eocene volcanic and intrusive rocks in Armenia and Azerbaijan, along the Lesser Caucasus Mountains (e.g. Allen and Armstrong, 2008), and plutons and volcanic rocks in Armenia (e.g. Moritz et al., 2016; Sahakyan et al., 2016), hampers calculations on additional $CO_2$ emissions within these regions.

Due to the absence of quantifications of the relation between the erupted volumes of volcanic rocks and emission of $CO_2$ in continental arcs, we make a comparison with the Deccan traps, for which this relation has been calculated. The Deccan traps have an estimated eruptive volume of volcanic and volcaniclastic rocks of $1.3*10^6$ $km^3$ (Jay and Widdowson, 2008), with an associated emission $4.14*10^{17}$ mol $CO_2$ (Tobin et al., 2017). From different estimates of volume and related $CO_2$ emissions of Tobin et al. (2017), we obtain a linear relation of lava volume (in $10^6$ $km^3$)/total $CO_2$ (in $10^{17}$ mol) $\approx 0.31$ for the Deccan traps.

$CO_2$ degassing rates for continental arcs may be similar to (Marty and Tolstikhin, 1998), or larger than for continental flood basalts (McKenzie et al., 2016; Wignall et al., 2009). As a conservative starting point, we assume a similar volume versus emission relationship as the Deccan traps, which implies a minimum estimate for $CO_2$ release from middle Eocene volcanism in Iran between $2.34*10^{16}$ and $1.22*10^{17}$ mol (see Table 2), which corresponds to 292-1461 Pg C. Moreover, the amount of $CO_2$ released during volcanic episodes has been shown to increase substantially if eruptions occur among carbonate-rich sediments (Lee et al., 2013; Lee and Lackey, 2015). For example, $CO_2$ released from carbonate sediments during the emplacement of the Emeishan large igneous province in the end-Guadalupian was estimated to be 3.6-8.6 times higher than the amount of $CO_2$ released by volcanic outgassing alone (Ganino and Arndt, 2009). Indeed, the Eocene volcanism in Iran erupted in shallow marine basins, and through significant amounts of carbonate-rich rocks of Jurassic, Cretaceous, and Paleogene age (e.g. Berberian and King, 1981). Glaus, (1965) mentions that middle Eocene limestones occur as lenticular masses within the basaltic flows, or as consistent horizons associated with tuffs. Verdel (2009) shows that Eocene volcanic rocks are formed in close association with Eocene limestones in north, west and east Iran. This is also the case in central Iran, which can be seen from geologic maps, such as the one from Qom (Emami, 1981). As a result, carbon release associated with the production of volcanic rocks in Iran could be much larger, potentially ranging from 1052 to 12,565 Pg C (see Table 2). This range of $CO_2$ emissions is compatible with the carbon cycle imbalance that drives the MECO in simple carbon cycle simulations constrained by available proxy data (roughly 2000-4000 Pg C; Henehan et al., 2020; Sluijs et al., 2013; van der Ploeg et al., 2018). Table 2 shows that middle Eocene volcanic rocks with thicknesses between 2 and 7 kilometres, and a contribution from limestones, give estimates that lie within the range expected for the MECO (marked in green in Table 2). There could have been a contribution to $CO_2$ through skarn formation by intrusive activity, which clusters around 40.5 Ma (see Figure 3D), although we currently lack the constraints to quantitatively assess this. Erosion has affected the entire Iranian plateau, and could have eroded away significant volumes of Eocene volcanic

rocks. Morley et al. (2009) and Ballato et al. (2011) note that clasts in the Lower and Upper Red formation (Oligocene-Miocene age), which in many places overlie Eocene volcanics, are for a large part made up of eroded Eocene volcanic rocks. Original thicknesses of Eocene volcanic rocks in Iran could thus have been larger, making our $CO_2$ output estimate a minimum estimate. Despite the fact that sampling biases (i.e. sampling is often focused on easily accessible sites and certain time periods) can never be avoided, our compilation of radiometric ages shows a good correlation to the geologic maps, in the sense that the radiometric ages confirm that the flare-up took place during the middle Eocene. We note that the Miocene peak (Figure 3C) is relatively high compared to the Eocene, which could be caused by a sampling bias, as the geologic maps (Sahandi et al., 2014) indicate that only 2-4% of Iran is covered by Miocene volcanic rocks.

## 5 Future perspectives

There are several obstacles in solidifying the link between warming during the MECO and volcanism in the Neotethys subduction zone. First of all, continental arcs are generally active for (tens of) millions of years, while the MECO has a duration of 500 kyr. Moreover, this duration is shorter than common uncertainties for radiometric ages in the Eocene, complicating the establishment of a causal relationship. This is important because a driver for the MECO requires excess $CO_2$ input only during the ~500 kyr spanning the MECO, and not during the time surrounding it (Sluijs et al., 2013). This is also supported by the drop in global ocean osmium isotope ratios, which is specifically associated with the MECO interval (van der Ploeg et al., 2018). Secondly, Iran is a relatively understudied area compared to other (continental) arcs. As a result of this, the amount of radiometric ages is low, with on average about 1 radiometric age for every several hundred $km^2$ of outcrop.

Therefore, the relation in time between the MECO and Neotethys arc flare-up calls for the development of much better age constraints of the volcanic deposits in Iran and this is certainly feasible. While most flare-ups have to be studied via their intrusive roots, as the extrusive record is removed through erosion (Ducea and Barton, 2007; De Silva et al., 2015), the extrusive record in Iran is extensive so that the ages can be mapped in high detail. Acquisition of radiometric ages throughout sections that cover the entire Eocene volcanic succession could aid in quantification of magmatic flux over time. Moreover, the respective roles of intrusive and extrusive rocks can be assessed to estimate the amount of volatiles of the igneous rocks, and sedimentological studies can provide minimum estimates on how much extrusive rock has been lost through erosion. Studies that focus on the interaction between carbonates and magma chambers could aid in quantifying the carbonate contribution to $CO_2$ release. This would help constrain $CO_2$ input rates across from the Neotethys flare-up to a narrower interval around the MECO.

## 6 Conclusions

We provide new Ar-Ar ages from volcanic rocks of the Azerbaijan-Bazman Arc in Iran and combine these with literature data to show that a flare-up of continental arc volcanism in Iran peaked about 40 Ma ago, conspicuously close to the Middle Eocene Climatic Optimum. We estimated volumes of middle Eocene volcanism in Iran to be between $7.6*10^4$ and $3.8*10^5$

km$^3$ . We compared the volume of middle Eocene volcanics in Iran to that of the Deccan traps and estimate that between 292
and 1461 Pg of carbon in the shape of $CO_2$ was released during deposition. Taking into account the fact that all volcanism
occurred in shallow marine basins and erupted in and through pre-existing carbonate-rich rocks, $CO_2$ release might have
been between 1052 and 12,565 Pg. Thicknesses of the middle Eocene volcanic succession between 2 and 7 kilometres, with
a contribution from carbonate-rich rocks result in estimates of released carbon that are in line with estimates for the MECO.
Although the flare-up must be dated much better to establish its chronological relation with the MECO in more detail, we
consider it a plausible major contributor to greenhouse warming during the MECO.

## 7 Supplementary materials

Examples of scans of thin sections are supplied in supplementary file S1. All details of literature ages and associated
references are added in supplementary files S2 and S3. S4 is a .kmz file that contains the GPS locations of the literature ages
(except of Shafaii Moghadam et al., (2020), who did not provide GPS locations), and can be opened in Google Earth. A
detailed description of Ar-Ar results per sample is provided in supplementary file S5. Supplementary files S6-S32 show the
results of the $^{40}Ar/^{39}Ar$ geochronology per experiment. S33 shows an extended version of the literature age plot of Figure 2C.

**Author contributions:** Fieldwork was undertaken by AvdB, MH and WK. AvdB, KFK and MH performed Ar-Ar dating.
Data analysis was performed by AvdB, KFK, RvdP, MJC and AS. All authors contributed to scientific discussions and were
involved in writing the manuscript. We thank two anonymous reviewers for their comments on an earlier version of this
manuscript, and three anonymous reviewers for their comments that have improved this manuscript.

**Competing interests:** The authors declare no competing interests.

**Acknowledgements**

This work was financially supported by Netherlands Organization for Scientific Research grant 865.10.011, awarded to WK,
and was carried out under the program of the Netherlands Earth System Science Centre, financially supported by the Dutch
Ministry of Education, Culture and Science. MLC and AS thank the Ammodo Foundation for funding unfettered research of
laureate AS. AS thanks the European Research Council for Consolidator Grant 771497 (SPANC). We thank Roel van Elsas
for help with Ar-Ar sample preparation.

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

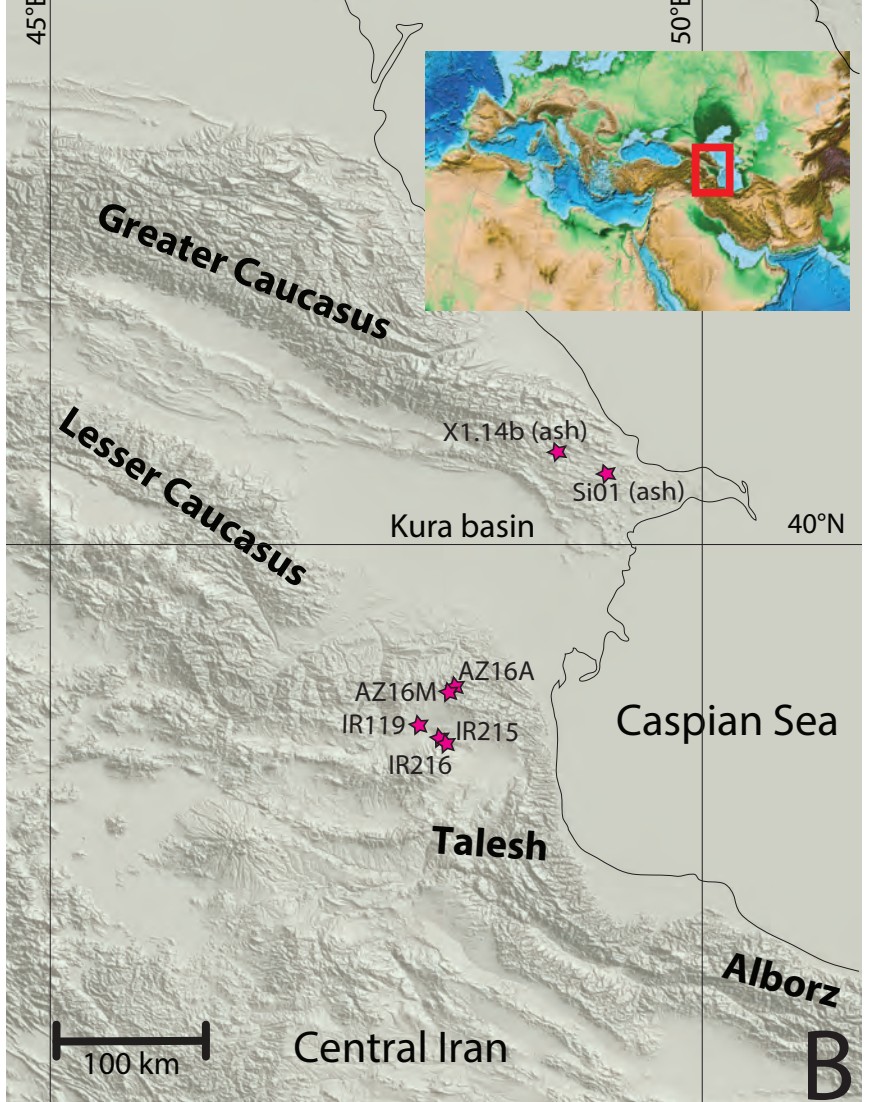

**Figure 1:**
**A.** Map showing the outcrop of Eocene volcanic rocks in Iran (modified after Agard et al., (2011) and shapefiles of Sahandi et al., (2014)). The five largest areas are shown with white outlines.
**B.** Sample locations of newly acquired $^{40}Ar/^{39}Ar$ ages.

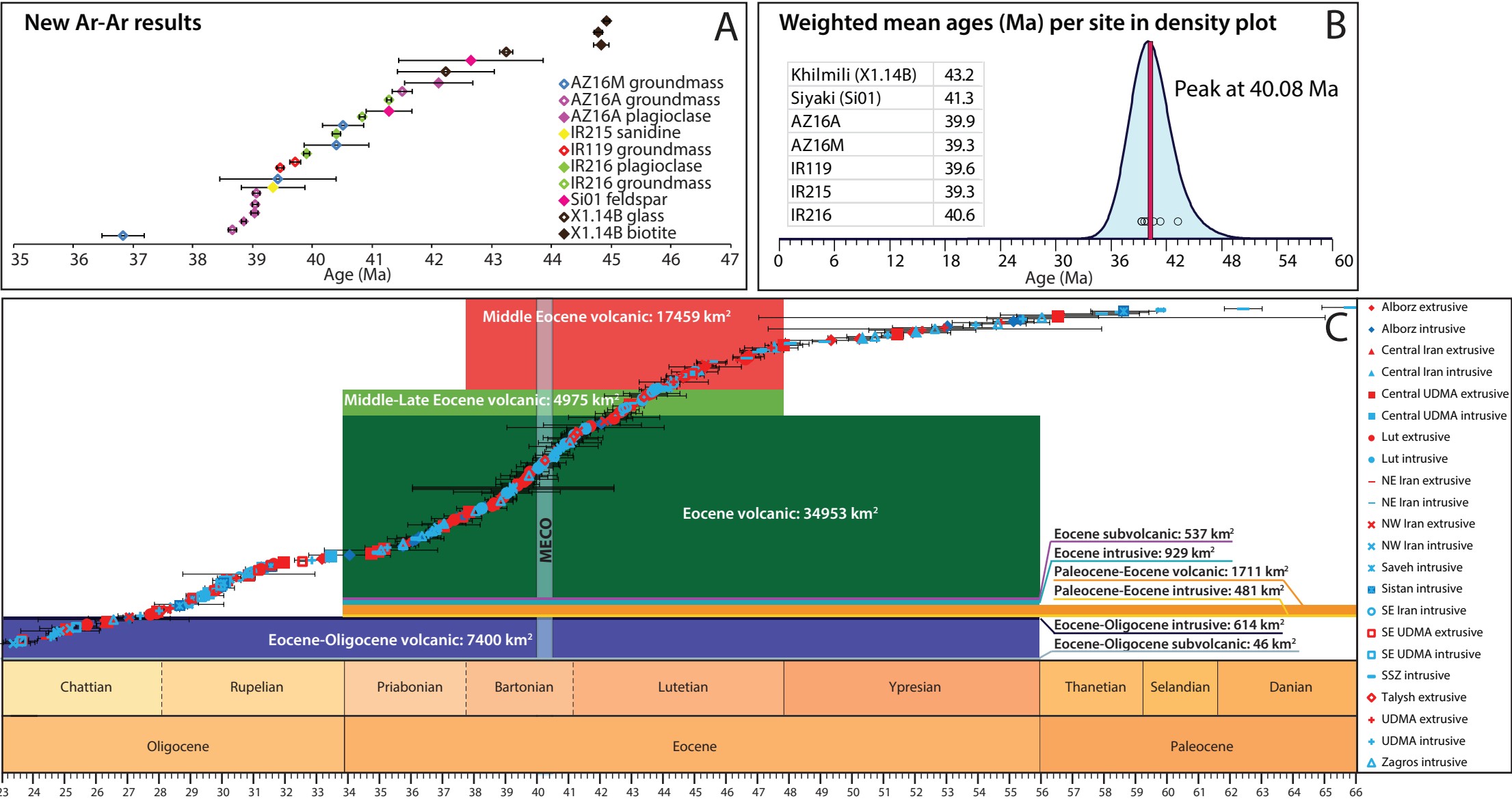

**Figure 2:**

**A.** $^{40}Ar/^{39}Ar$ dates of rocks from northwest Iran, south Azerbaijan and the Kura basin with uncertainties (1σ).

**B.** Kernel density plot in blue (Vermeesch, 2012) of combined $^{40}Ar/^{39}Ar$ ages. The duration and timing of the MECO event is indicated by the pink band.

**C.** Timescale (created with TSCreator) with scaled eruptive areas (from Sahandi et al., 2014), color legend for areas is the same as in figure 1A. Also plotted are radiometric ages from literature with associated 1σ errors, sorted by age, Y-axis is arbitrary unit. Red markers represent extrusive ages, blue markers represent intrusive ages.

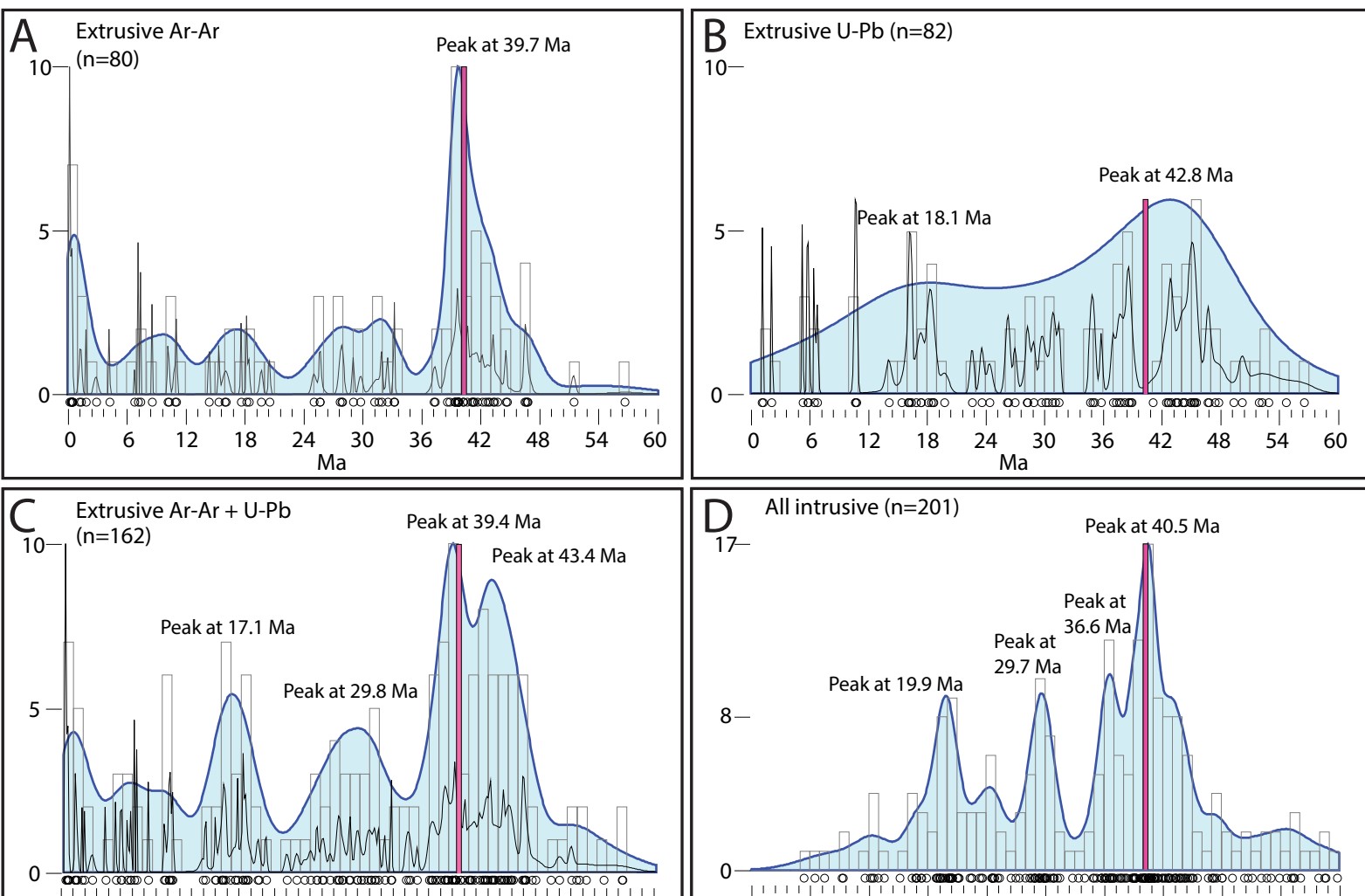

**Figure 3:**
Radio-isotope ages from 0-60 Ma, compiled from literature, combined with our newly obtained ages.
Black thin line represents the probability density plot (PDP), blue filled line represents the kernel density estimate (KDE), boxes represent histograms (numbers on Y-axis). The timing and duration of the MECO is indicated by the pink box.
**A.** $^{40}$Ar/$^{39}$Ar ages from literature combined with newly obtained ages for extrusive rocks only.
**B.** U-Pb ages from literature for extrusive rocks only.
**C.** $^{40}$Ar/$^{39}$Ar data and U-Pb data from extrusive rocks only.
**D.** Combined ages from literature for intrusive rocks only.

| Region | Section | Formation | Type | Sample | Lat | Long | Areal extent on shapefile (km$^2$) | Mineral | Excel file | Age plateau | Error (2σ) |
|---|---|---|---|---|---|---|---|---|---|---|---|
| Kura basin (Azerbaijan) | Khilmili | Koun | Ash | X1.14B | 40.68640 | 48.87632 | N/A | glass | VU101C-A4a | 42.23 | 1.62 |
| | | | | | | | | | VU101B-A4e | 43.24 | 0.20 |
| | | | | | | | | biotite | VU101B-A8a | 44.83 | 0.25 |
| | | | | | | | | | VU101B-A8b | 44.78 | 0.13 |
| | | | | | | | | | VU101B-A8e | 44.92 | 0.11 |
| Kura basin (Azerbaijan) | Siyaki | Koun | Ash | Si01 | 40.54582 | 49.25776 | N/A | feldspar | VU101B-A7a | 42.65 | 2.43 |
| | | | | | | | | | VU101B-A7e | 41.28 | 0.78 |
| Talysh (Azerbaijan) | | Peshtasar | Basalt | AZ16A | 38.86672 | 48.05018 | N/A | groundmass | VU101B-A1c | 41.50 | 0.35 |
| | | | | | | | | | VU101B-A1da | 39.06 | 0.12 |
| | | | | | | | | | VU101B-A1db | 39.04 | 0.11 |
| | | | | | | | | | VU101B-A1f | 38.66 | 0.16 |
| | | | | | | | | plagioclase | VU101B-A3c | 42.11 | 1.12 |
| | | | | | | | | | VU101B-A3da | 39.03 | 0.06 |
| | | | | | | | | | VU101B-A3db | 38.85 | 0.10 |
| Talysh (Azerbaijan) | | Peshtasar | Basalt | AZ16M | 38.90105 | 48.09660 | N/A | groundmass | VU101B-A2c | 40.51 | 0.68 |
| | | | | | | | | | VU101B-A2da | 36.83 | 0.69 |
| | | | | | | | | | VU101B-A2db | 39.42 | 1.95 |
| | | | | | | | | | VU101B-A2f | 40.40 | 1.06 |
| Talesh (Iran) | | Karaj | Trachyte | IR119 | 38.62625 | 47.77411 | 168 | groundmass | VU107-A2_1 | 39.71 | 0.09 |
| | | | | | | | | | VU107-A2_2 | 39.46 | 0.12 |
| Talesh (Iran) | | Karaj | Trachyandesite | IR215 | 38.42648 | 47.95920 | 406 | feldspar | VU107-A4_1 | 39.34 | 1.08 |
| Talesh (Iran) | | Karaj | Tracyandesite | IR216 | 38.40969 | 47.98619 | 406 | plagioclase | VU107-A5_1 | 40.40 | 0.13 |
| | | | | | | | | | VU107-A5_2 | 39.90 | 0.11 |
| | | | | | | | | groundmass | VU107-A6_1 | 41.28 | 0.10 |
| | | | | | | | | | VU107-A6_2 | 40.83 | 0.11 |
| Western Alborz (Iran) | | Karaj | Trachybasalt | IR22 | 36.48612 | 48.98974 | 281 | groundmass | VU107-A1_1 | 35.98 | 1.11 |
| | | | | | | | | | VU107-A1_2 | 51.64 | 1.26 |

**Table 1:** Details of samples for Ar-Ar dating.

| Region | Age | Area (km²) | Thickness (km) | Volume (km³) | vol Iran/Deccan | CO₂ Iran (mol) | CO₂ (g) | CO₂ (pg) | C (pg) | CO₂ limestone multiplier | CO₂ with lst (pg) | C with lst (pg) |
|---|---|---|---|---|---|---|---|---|---|---|---|---|
| Iran | Middle Eocene | 38223 | 2 | 76446 | 0.06 | 2.43E+16 | 1.07E+18 | 1071 | 292 | 3.6 | 3857 | 1052 |
|  |  |  | 2 | 76446 | 0.06 | 2.43E+16 | 1.07E+18 | 1071 | 292 | 8.6 | 9214 | 2513 |
|  |  |  | 3 | 114669 | 0.09 | 3.65E+16 | 1.61E+18 | 1607 | 438 | 3.6 | 5786 | 1578 |
|  |  |  | 3 | 114669 | 0.09 | 3.65E+16 | 1.61E+18 | 1607 | 438 | 8.6 | 13821 | 3769 |
|  |  |  | 4 | 152892 | 0.12 | 4.87E+16 | 2.14E+18 | 2143 | 584 | 3.6 | 7714 | 2104 |
|  |  |  | 4 | 152892 | 0.12 | 4.87E+16 | 2.14E+18 | 2143 | 584 | 8.6 | 18429 | 5026 |
|  |  |  | 5 | 191115 | 0.15 | 6.09E+16 | 2.68E+18 | 2679 | 731 | 3.6 | 9643 | 2630 |
|  |  |  | 5 | 191115 | 0.15 | 6.09E+16 | 2.68E+18 | 2679 | 731 | 8.6 | 23036 | 6282 |
|  |  |  | 6 | 229338 | 0.18 | 7.30E+16 | 3.21E+18 | 3214 | 877 | 3.6 | 11571 | 3156 |
|  |  |  | 6 | 229338 | 0.18 | 7.30E+16 | 3.21E+18 | 3214 | 877 | 8.6 | 27643 | 7539 |
|  |  |  | 7 | 267561 | 0.21 | 8.52E+16 | 3.75E+18 | 3750 | 1023 | 3.6 | 13500 | 3682 |
|  |  |  | 7 | 267561 | 0.21 | 8.52E+16 | 3.75E+18 | 3750 | 1023 | 8.6 | 32250 | 8795 |
|  |  |  | 8 | 305784 | 0.24 | 9.74E+16 | 4.29E+18 | 4286 | 1169 | 3.6 | 15429 | 4208 |
|  |  |  | 8 | 305784 | 0.24 | 9.74E+16 | 4.29E+18 | 4286 | 1169 | 8.6 | 36857 | 10052 |
|  |  |  | 9 | 344007 | 0.26 | 1.10E+17 | 4.82E+18 | 4821 | 1315 | 3.6 | 17357 | 4734 |
|  |  |  | 9 | 344007 | 0.26 | 1.10E+17 | 4.82E+18 | 4821 | 1315 | 8.6 | 41464 | 11308 |
|  |  |  | 10 | 382230 | 0.29 | 1.22E+17 | 5.36E+18 | 5357 | 1461 | 3.6 | 19286 | 5260 |
|  |  |  | 10 | 382230 | 0.29 | 1.22E+17 | 5.36E+18 | 5357 | 1461 | 8.6 | 46071 | 12565 |

**Table 2:** Estimates for the amount of released carbon for different volumes of middle Eocene volcanic rocks in Iran. Estimates marked in green are within estimates for the MECO (2000-4000 Pg C; Henehan et al., 2020; Sluijs et al., 2013; van der Ploeg et al., 2018).