# Peer review of "Exploring a link between the Middle Eocene Climatic Optimum and Neotethys continental arc flare-up"

_Climate of the Past, 2020_

## Referee Comment (RC1) · Anonymous Referee #1 · 27 Apr 2020

This is an interesting test between the possible link of arc volcanism and climate change. It fits the scope of the journal. After the carefully read, I found that the ms has many logical and method flaws, which needs significant revision.

The successful link between the arc volcanism and climate change depends on how much carbon dioxide has been outputted through the ∼40 million years old volcanos. Firstly, the authors claimed there is an intensive eruption pulse at 40 million years based on their own and published data. However, the crucial point is how much 40 Ma volcanism has erupted. The assumption of the authors is improper and geologically impossible. 1-The authors assumed the total area of the 40 Ma is 40,000 km2, and

the thickness is 3-9km, so the volume of the middle Eocene volcano is 100,000-350 000 km3. 9 km is almost the whole thickness of the upper crust, so how could one volcanic eruption make 1/3 of the crust. After I checked the reference Verdel et al., 2011, they claimed the whole Paleogene (66-23 Ma) strata, including the volcanism and sedimentary rocks in the UDMA is 3-9 km. Clearly, the authors have much overestimated the thickness of 40 Ma volcanic rocks. According to figure 2, we see volcanic events throughout the whole Eocene. Although there is an intense event at 40 Ma, still, the 40 Ma volcanic rocks are only a part of the Paleogene volcanic strata (3-9 km). You must be precise how thick is the ∼40 Ma rock.

2-The second point is that the authors probably underestimated CO2 output based on their calculation.

The authors compared the size of the arc volcanism with the large igneous province in Deccan and directly used the CO2 output data from LIPs. However, the compositions of arc volcanism are fundamentally different from those of LIP. The arc volcanism is more felsic that is compared to the dominated basalt of LIPs. Then the arc volcanism is much enriched with volatile like carbon (0.6-1.3 wt%, Wallace et al.,2005), water (4 wt%, Plank et al.,2013.). Therefore, if the authors used the arc data, I think the output of carbon maybe more. Because of the compositional difference, the felsic arc volcanism is more like to interact with the carbonate to form skarn that further releases more CO2. The LIP basalts are more likely to assimilate with carbonate and related to fewer CO2 (Carter et al., 2016). On the contrary, the basalts are much easier to weathering, which consumes many CO2, which may cause cooling.

3-Current data do not support their conclusion. The authors must recalculate the budget. MECO is a global effect. I suggest the authors also add some discussion on the possible Eocene arc volcanism at other places like along the Tethyan region and the Cordillera region in the eastern Pacific. As far as I know, the post-Laramide volcanism is also very strong.

---

## Referee Comment (RC2) · Anonymous Referee #2 · 13 May 2020

Detailed comments on the manuscript of "Exploring a link between the Middle Eocene Climatic Optimum and Neotethys continental arc flare-up" have been made as follows.

This paper presents new data, idea and explanation about a link between the Middle Eocene Climatic Optimum and Neotethys continental arc flare-up. It is sure that this interpretation in this paper presented will therefore be of considerable helpful for anyone working in this field. I fully support publication of this work, and the comments that I have listed below are chiefly intended to help the authors make their manuscript as clear and accessible to potential readers as possible.

I suggest that the author may consider adding a new section of "Geological background". The authors may briefly review all previous studies and ideas partly concerning with the relation between petrogenesis and tectonic evolution history based on clearly and strongly geological evidence because conflicting data and hypotheses concerning about geological history and petrogenesis in the studied area have been presented in previous studies. I think that if there is the description about the geological outline, which is also ok although it seems a little simple. Importantly, magmatism (including volcanism) with different characteristics in geochemical composition, mantle source regions and geodynamic setting would have full differences in eruptive column heights for volcanism only, volatile (including $CO_2$) degassing rates and fluxes, and amounts of outgassing gases from magmatic activities, which are importantly controlling parameters on climate changes related to magmatism (including volcanism). If calculated and/or analysed results of the parameters (including the eruptive column heights for volcanism only, volatile (including $CO_2$) degassing rates and fluxes, and amounts of outgassing gases from magmatic activities) cannot be well determined by the magmatic (including volcanic) bodies themselves based on the melt inclusion sample analysis in the lab (Including EMP, Raman, SIMES, etc.), instead of comparison with those released from other volcanic activities (e.g. the Deccan traps in this paper), the final results and even conclusions of which would possibly need to reevaluated, because it is not easy to develop a link in these parameters (including the eruptive column heights for volcanism only, volatile (including $CO_2$) degassing rates and fluxes, and amounts of outgassing gases from magmatic activities) between magmatism (including volcanism) with different characteristics in geochemical composition, mantle source regions and geodynamic setting.

I suggest that the author may further explain the petrologic reason, rationale and geochemical basis of the comparisons in magmatic $CO_2$ outgassing rate (or amount) between the Deccan traps and magmatic activities in this paper (see details in about Line 140), which may be thought to be an potentially estimated method of the magmatic $CO_2$ outgassing rate (or amount).

Additionally, it should really be pointed out here that magmatism concerned with in this paper belongs to HKCA volcanism, which is related to oceanic plate subduction. But, many previous studies (including a recent study published in Geology-2019) indicate that this kind of HKCA volcanism may act as a key driver of the late Paleozoic ice age (Soreghan, G.S., Soreghan, M.J., and Heavens, N.G., 2019, Explosive volcanism as a key driver of the late Paleozoic ice age: Geology.). Thus, magmatism with similar geodynamic setting may have total different the magmatic $CO_2$ outgassing rate (or amount), which are very comment situations. However, the Deccan traps and magmatic activities in this paper have totally different geodynamic settings, thus i hope the author may further explain the reason of the comparisons in magmatic $CO_2$ outgassing rate (or amount) between the Deccan traps and magmatic activities in this paper (see details in about Line 140). Whether or not are the results from the comparisons in this paper better than those in previous studies (Including EMP, Raman, SIMES, etc.)?

---

## Author Comment (AC1) · 22 Jun 2020

We thank the two anonymous reviewers for their helpful comments on our manuscript, and provide a response to each of their comments below.

[Author reply to comment 1 of Reviewer 1]: The key point of our response to this comment is that there is no one single eruption around 40 Ma. This was perhaps not stated clearly enough in the manuscript, so we will rephrase this. We do not state that all these volcanic rocks have erupted in a single volcanic eruption as the reviewer seems to imply. Rather, we see a large increase in volcanic activity all around Iran in the middle Eocene, and this activity is observed in all different regions. Consequently, there

must have been many volcanic eruptions that all together contributed to the thickness of (middle) Eocene volcanic units in Iran. We further emphasise that the thicknesses that we report are fully in line with the statement of Verdel et al. (2011): "Reported thicknesses of Paleogene volcanic and sedimentary rocks are âĹij3–9 km in the Uru-mieh‐Dokhtar belt (Figure 1) in central Iran and the Alborz Mountains in northern Iran [e.g., Förster et al., 1972; Annells et al., 1975; Hassanzadeh, 1993; Morley et al., 2009]." This is also clear from other literature on this topic. Berberian & King (1981) state that "Extensive volcanism, with a wide range of composition, started in the Eocene Period (50 Ma) and continued for the rest of the period with the climax in Middle Eocene time (about 47-42 Ma). Despite their great thickness (locally up to 6 and 12 km) and wide distribution, the volcanics and tuffs were formed within a relatively short time interval." Stöcklin (1974) mentions Eocene volcanic rocks in the Alborz to have a thickness of 3-5 km. Allen et al. (2003) mention a thickness of 5 km for the Eocene Karaj Formation (which consists mostly of volcanic and volcaniclastic rocks) in the Alborz. Taking into account all these different estimates, we feel that the values of 3-9 km that we use in our calculations are reasonable and agree well with estimates from literature.

To clarify this issue in the text, we will add the above references to support the statement on the thickness of the volcanic deposits. Moreover, we will clarify that we do not intend to suggest that one eruption caused all these deposits but that they rather represent a phase of intensified volcanism.

[Author reply to comment 2 of Reviewer 1]: We fully agree with the reviewer that our estimates are likely underestimates. We deliberately chose a conservative approach. Therefore, we mention in line 139-140 that our assumption of a similar volume versus emission relationship as the Deccan traps results in a minimum estimate of CO2. We will put more emphasis on our estimates being minimum estimates, and the carbon contribution by Eocene volcanism in Iran could have been much larger (see also our response to the next comment).

We will add the following part to the discussion: "Erosion has affected the entire Iranian plateau, and could have eroded away significant volumes of Eocene volcanic rocks. Morley et al. (2009) and Ballato et al. (2011) note that clasts in the Lower and Upper Red formation (Oligocene-Miocene age), which in many places overlie Eocene volcanics, are for a large part made up of eroded Eocene volcanic rocks. Original thicknesses of Eocene volcanic rocks in Iran could thus have been larger, making our $CO_2$ output estimate a minimum estimate."

[Author reply to comment 3 of Reviewer 1]: We agree with the reviewer that other volcanically active areas in the Tethyan region might have played a role during the MECO (Armenia, Georgia and Turkey) as we have mentioned in lines 46-48. In addition, we will add the following part to the discussion: "Our estimates of $CO_2$ release due to middle Eocene volcanism in Iran are likely underestimates, as there is volcanism in other regions along the Neotethys subduction zone. Unfortunately, the lack of shapefiles of Eocene volcanic and intrusive rocks in Armenia and Azerbaijan, along the Lesser Caucasus Mountains (e.g. Allen and Armstrong, 2008), and plutons and volcanic rocks in Armenia (e.g. Moritz et al., 2016; Sahakyan et al., 2016), hampers calculations on additional $CO_2$ emissions within these regions."

We have done a thorough literature study of other inferred causes of the MECO around the globe, such as increased mid-ocean ridge volcanism (Bohaty et al., 2009), increased metamorphic decarbonation associated with Himalayan uplift (Kerrick and Caldeira, 1999; Pearson, 2010), increased extrusive arc volcanism in the Pacific rim (Cambray and Cadet, 1996), increased carbonatite magmatism in the East African Rift (Bailey, 1992, 1993), or increased Cordilleran belt volcanism (Kerrick and Caldeira, 1998). However, we did not find confirming radiometric ages or other evidence that indicated that other regions showed a temporal link to the MECO event. We thus decided to focus on our own data, instead of discussing the absence of evidence from other regions.

---

## Author Response (AR1)

**Response to reviewers:**

*We thank the two anonymous reviewers for their helpful comments on our manuscript, and provide a response to each of their comments below.*

**Reviewer 1**
This is an interesting test between the possible link of arc volcanism and climate change. It fits the scope of the journal. After the carefully read, I found that the ms has many logical and method flaws, which needs significant revision. The successful link between the arc volcanism and climate change depends on how much carbon dioxide has been outputted through the 40 million years old volcanos. Firstly, the authors claimed there is an intensive eruption pulse at 40 million years based on their own and published data. However, the crucial point is how much 40 Ma volcanism has erupted. The assumption of the authors is improper and geologically impossible.

1-The authors assumed the total area of the 40 Ma is 40,000 km2, and the thickness is 3-9km, so the volume of the middle Eocene volcano is 100,000-350,000 km. 3. 9 km is almost the whole thickness of the upper crust, so how could one volcanic eruption make 1/3 of the crust. After I checked the reference Verdel et al., 2011, they claimed the whole Paleogene (66-23 Ma) strata, including the volcanism and sedimentary rocks in the UDMA is 3-9 km. Clearly, the authors have much overestimated the thickness of 40 Ma volcanic rocks. According to figure 2, we see volcanic events throughout the whole Eocene. Although there is an intense event at 40 Ma, still, the 40 Ma volcanic rocks are only a part of the Paleogene volcanic strata (3-9 km). You must be precise how thick is the 40 Ma rock.

*The key point of our response to this comment is that there is no one single eruption around 40 Ma. This was perhaps not stated clearly enough in the manuscript, so we have put more emphasis on this. We do not state that all these volcanic rocks have erupted in a single volcanic eruption as the reviewer seems to imply. Rather, we see a large increase in volcanic activity all around Iran in the middle Eocene, and this activity is observed in all different regions. Consequently, there must have been many volcanic eruptions that all together contributed to the thickness of (middle) Eocene volcanic units in Iran. We further emphasise that the thicknesses that we report are fully in line with the statement of Verdel et al. (2011): "Reported thicknesses of Paleogene volcanic and sedimentary rocks are ~3–9 km in the Urumieh-Dokhtar belt (Figure 1) in central Iran and the Alborz Mountains in northern Iran [e.g., Förster et al., 1972; Annells et al., 1975; Hassanzadeh, 1993; Morley et al., 2009]." This is also clear from other literature on this topic. Berberian & King (1981) state that "Extensive volcanism, with a wide range of composition, started in the Eocene Period (50 Ma) and continued for the rest of the period with the climax in Middle Eocene time (about 47-42 Ma). Despite their great thickness (locally up to 6 and 12 km) and wide distribution, the volcanics and tuffs were formed within a relatively short time interval." Stöcklin (1974) mentions Eocene volcanic rocks in the Alborz to have a thickness of 3-5 km. Allen et al. (2003) mention a thickness of 5 km for the Eocene Karaj Formation (which consists mostly of volcanic and volcaniclastic rocks) in the Alborz. Taking into account all these different estimates, we feel that the values of 3-9 km that we use in our calculations are reasonable and agree well with estimates from literature.*

*To clarify this issue in the text, we have added the above references to support the statement on the thickness of the volcanic deposits. We have modified line 136-139 to: "In the Alborz and Central Iran, middle Eocene extrusive volcanic formations are reported to be very thick, with estimates ranging from 3-5 km in the Alborz Mountains (Stöcklin, 1974), to 6-12 km locally throughout nearly all of Iran (Berberian and King, 1981). More recent estimates of the thickness are 3-9 kilometers (e.g. Morley et al., 2009; Verdel et al., 2011)."*

*Moreover, we have clarified that we do not intend to suggest that one eruption caused all these deposits but that they rather represent a phase of intensified volcanism, by adding (line 136): "These volcanic rocks were produced by numerous eruptions throughout the middle Eocene."*

2-The second point is that the authors probably underestimated CO2 output based on their calculation. The authors compared the size of the arc volcanism with the large igneous province in Deccan and directly used the CO2 output data from LIPs. However, the compositions of arc volcanism are fundamentally different from those of LIP. The arc volcanism is more felsic that is compared to the dominated basalt of LIPs. Then the arc volcanism is much enriched with volatile like carbon (0.6-1.3 wt%, Wallace et al.,2005), water (4 wt%, Plank et al.,2013.). Therefore, if the authors used the arc data, I think the output of carbon maybe more. Because of the compositional difference, the felsic arc volcanism is more like to interact with the carbonate to form skarn that further releases more CO2. The LIP basalts are more likely to assimilate with carbonate and related to fewer CO2 (Carter et al., 2016). On the contrary, the basalts are much easier to weathering, which consumes many CO2, which may cause cooling.

*We fully agree with the reviewer that our estimates are likely underestimates. We deliberately chose a conservative approach. Therefore, we mention in line 139-140 that our assumption of a similar volume versus emission relationship as the Deccan traps results in a minimum estimate of $CO_2$. We will put more emphasis on our estimates being minimum estimates, and the carbon contribution by Eocene volcanism in Iran could have been much larger (see also our response to the next comment).*

*We have added the following part to the discussion (lines 171-175): "Erosion has affected the entire Iranian plateau, and could have eroded away significant volumes of Eocene volcanic rocks. Morley et al. (2009) and Ballato et al. (2011) note that clasts in the Lower and Upper Red formation (Oligocene-Miocene age), which in many places overlie Eocene volcanics, are for a large part made up of eroded Eocene volcanic rocks. Original thicknesses of Eocene volcanic rocks in Iran could thus have been larger, making our $CO_2$ output estimate a minimum estimate."*

3-Current data do not support their conclusion. The authors must recalculate the budget. MECO is a global effect. I suggest the authors also add some discussion on the possible Eocene arc volcanism at other places like along the Tethyan region and the Cordillera region in the eastern Pacific. As far as I know, the post-Laramide volcanism is also very strong.

*We agree with the reviewer that other volcanically active areas in the Tethyan region might have played a role during the MECO (Armenia, Georgia and Turkey) as we have mentioned in lines 46-48 (in revised manuscript lines 54-56). In addition, we have added the following part to the discussion (lines 141-145): "Our estimates of $CO_2$ release due to middle Eocene volcanism in Iran are likely underestimates, as there is volcanism in other regions along the Neotethys subduction zone. Unfortunately, the lack of shapefiles of Eocene volcanic and intrusive rocks in Armenia and Azerbaijan, along the Lesser Caucasus Mountains (e.g. Allen and Armstrong, 2008), and plutons and volcanic rocks in Armenia (e.g. Moritz et al., 2016; Sahakyan et al., 2016), hampers calculations on additional $CO_2$ emissions within these regions."*

*We have done a thorough literature study of other inferred causes of the MECO around the globe, such as increased mid-ocean ridge volcanism (Bohaty et al., 2009), increased metamorphic decarbonation associated with Himalayan uplift (Kerrick and Caldeira, 1999; Pearson, 2010), increased extrusive arc volcanism in the Pacific rim (Cambray and Cadet, 1996), or increased carbonatite magmatism in the East African Rift (Bailey, 1992, 1993), or increased Cordilleran belt volcanism (Kerrick and Caldeira, 1998). However, we did not find confirming radiometric ages or*

*other evidence that indicated that other regions showed a temporal link to the MECO event. We thus decided to focus on our own data, instead of discussing the absence of evidence from other regions.*

**Reviewer 2**
Detailed comments on the manuscript of "Exploring a link between the Middle Eocene Climatic Optimum and Neotethys continental arc flare-up" have been made as follows. This paper presents new data, idea and explanation about a link between the Middle Eocene Climatic Optimum and Neotethys continental arc flare-up. It is sure that this interpretation in this paper presented will therefore be of considerable helpful for anyone working in this field. I fully support publication of this work, and the comments that I have listed below are chiefly intended to help the authors make their manuscript as clear and accessible to potential readers as possible.

*We thank the reviewer for their kind words and support of our manuscript.*

I suggest that the author may consider adding a new section of "Geological background". The authors may briefly review all previous studies and ideas partly concerning with the relation between petrogenesis and tectonic evolution history based on clearly and strongly geological evidence because conflicting data and hypotheses concerning about geological history and petrogenesis in the studied area have been presented in previous studies. I think that if there is the description about the geological outline, which is also ok although it seems a little simple.

*We agree with the reviewer there are many different and conflicting tectonic and petrogenic models for Eocene volcanism in Iran. A thorough review of all of the geologic settings of these different areas of Iran is a huge task that is deserving of a study in its own right. We mainly intend to show in this study that there is a huge increase in volcanism in Iran during the Eocene in all of these regions, regardless of their tectonic history and petrogenesis, which is why we do not discuss all the petrologic models in detail. To give some more background information, we have added to the Introduction (lines 46-53):*
*"Geologic settings of the Eocene volcanic regions in Iran differ. Extensive magmatism in the Lut block is regarded by Pang et al. (2013) to be the result of post-collisional convective removal of the lithosphere and not directly related to subduction. Volcanism in the Sabzevar zone is linked by Moghadam et al. (2016) to lithospheric delamination, possibly assisted by slab-breakoff. In the Talesh/Alborz region, there are conflicting theories on the formation of the volcanic rocks. Asiabanha & Foden (2012) mention a post-collisional transition to a continental arc in their title, but then describe the volcanism as back-arc volcanism. Van der Boon (2017) gives an overview of proposed conflicting settings for volcanism in the Alborz. It is striking that in most of the areas in Iran, the flare-up is linked to an extensional setting (e.g. Verdel et al., 2011), which makes it different from other flare-ups (e.g. Ducea et al., 2015; Ducea and Barton, 2007)."*

Importantly, magmatism (including volcanism) with different characteristics in geochemical composition, mantle source regions and geodynamic setting would have full differences in eruptive column heights for volcanism only, volatile (including $CO_2$) degassing rates and fluxes, and amounts of outgassing gases from magmatic activities, which are importantly controlling parameters on climate changes related to magmatism (including volcanism). If calculated and/or analysed results of the parameters (including the eruptive column heights for volcanism only, volatile (including $CO_2$) degassing rates and fluxes, and amounts of outgassing gases from magmatic activities) cannot be well determined by the magmatic (including volcanic) bodies themselves based on the melt inclusion sample analysis in the lab (Including EMP, Raman, SIMES, etc.), instead of comparison with those released from other volcanic activities (e.g. the Deccan traps in this paper), the final results and even conclusions of which would possibly need to reevaluated, because it is not easy to develop a link in these parameters (including the eruptive column heights for volcanism only, volatile (including $CO_2$)

degassing rates and fluxes, and amounts of outgassing gases from magmatic activities) between magmatism (including volcanism) with different characteristics in geochemical composition, mantle source regions and geodynamic setting. I suggest that the author may further explain the petrologic reason, rationale and geochemical basis of the comparisons in magmatic CO2 outgassing rate (or amount) between the Deccan traps and magmatic activities in this paper (see details in about Line 140), which may be thought to be an potentially estimated method of the magmatic CO2 outgassing rate (or amount).

*We fully agree with the reviewer that more detailed research on this topic could strengthen or invalidate our results, and we hope that our study encourages further study on the Iranian Eocene volcanics and their $CO_2$ emissions. Here we describe the state-of-the-art regarding the dating of the volcanic deposits. There is currently not a lot of data available on Eocene melt inclusions in Iran, there are only very few that are focused on mineralisation, so this kind of work could provide more insights into settings of Eocene volcanism, ideally on a similar large scale as we present our dating.*

*In order to bridge the gap between the scales, we thus have to rely on the scarce information that is available on magmatic volumes and related $CO_2$ content, and only the well-studied Deccan traps have estimates for this. We thus use what is available, and that is unfortunately only information from the Deccan traps. To our knowledge, there have been no studies that constrained the amount of $CO_2$ per volume of arc volcanic rocks. We note that that is also a more difficult task, due to the varied nature of the different rock types in arcs (i.e. nearly every type from mafic to felsic, while LIPs consist mainly of basalt).*

*To comply with the reviewer's comment, we have modified lines 132-133 (in revised manuscript lines 146-147) to: "Due to the absence of quantifications of the relation between the erupted volumes of volcanic rocks and emission of $CO_2$ in continental arcs, we make a comparison with the Deccan traps, for which this relation has been calculated." As mentioned in lines 139-140 (in revised manuscript lines 153-154), this likely results in a minimum estimate for the amount of $CO_2$ related to Eocene volcanic activity in Iran.*

Additionally, it should really be pointed out here that magmatism concerned with in this paper belongs to HKCA volcanism, which is related to oceanic plate subduction. But, many previous studies (including a recent study published in Geology-2019) indicate that this kind of HKCA volcanism may act as a key driver of the late Paleozoic ice age (Soreghan, G.S., Soreghan, M.J., and Heavens, N.G., 2019, Explosive volcanism as a key driver of the late Paleozoic ice age: Geology.). Thus, magmatism with similar geodynamic setting may have total different the magmatic CO2 outgassing rate (or amount), which are very comment situations.

*The study of Soreghan et al. 2019 is very intriguing but at the same time highly speculative. For example,* Lee and Dee (2019) *discuss the Soreghan et al. paper, and state that individual eruptions might manifest as short-term cooling events superimposed on an otherwise warmer baseline. This is more consistent with the paradigm.*
*The Eocene in Iran consists of many units that contain volcaniclastic rocks that have been interpreted as the result of explosive eruptions that might potentially cause some degree of dimming (e.g.* Asiabanha et al., 2012; Asiabanha and Bardintzeff, 2013). *Many of the Eocene volcanic units in Iran are mapped as 'Eocene volcanics' and which doesn't allow us to precisely quantify the amount of pyroclastics and ignimbrites, as Soreghan et al. (2019) have done. Also eruption magnitudes are not estimated for Eocene volcanic rocks in Iran, and there have been no reports of large calderas besides one in Tafresh (Ghorbani & Bezenjani, 2011).*

*Most importantly, however, we here test for a link between a phase of global warming through volcanic CO$_2$ forcing rather than a cooling through volcanic aerosol formation. For these reasons, we at this point choose not to discuss this issue.*

However, the Deccan traps and magmatic activities in this paper have totally different geodynamic settings, thus i hope the author may further explain the reason of the comparisons in magmatic CO2 outgassing rate (or amount) between the Deccan traps and magmatic activities in this paper (see details in about Line 140). Whether or not are the results from the comparisons in this paper better than those in previous studies (Including EMP, Raman, SIMES, etc.)?

*Please see our response to a similar comment above.*

Other changes:

-In Figure 3, we moved the position of the labels of the peaks (e.g. Peak at 29.5 Ma) slightly to improve readability as it was overlapping with the graph in the first version.
-Removed 'extrusive' in line 18,136 and 138 as that is a pleonasm when combined with 'volcanic'.
-Added reference Boscolo-Galazzo et al. 2013 in the introduction (line 28).

[revised manuscript text omitted]

---

## Referee Report (RR1)

I have read through the updated version of "Exploring a link between the Middle Eocene Climatic Optimum and Neotethys continental arc flare-up" by Van der Boon et al., and the response to my original comments. I recognize that the authors have made a good faith effort to address my original concerns and that some assumptions are now better motivated (especially on the thickness of the Eocene rocks). I still disagree with some answers and observations. Please see below:

- The authors write as reply to my comment:
*"Furthermore, Verdel (2009) shows in Figure 5 of Chapter 3 that most of the flare-up is during the middle Eocene for North, West and East Iran, and only in Central Iran extends also into the lower Eocene."*
Figure of Verdel's thesis shows 3 ages for Central Iran 52.9±3.3, 50.0±4.4, 54.7±3.1, and one age for northern Iran 52.2±3.4 Ma; all these ages fall within the error, so it is not clear to me what is the age of the Karaj Formation (maybe we should look at paleontological works on the Ziarat Formation). In any case the main point is to have an increase in magmatic flux around 42-40 Ma. Intense magmatism between 50 and 45 Ma or 55 and 45 Ma doesn't help. To have an impact on MECO should be coeval and I guess that ,despite all difficulties, your data seems to go in that direction.

- The authors write in the main text:
*"Field studies have often suggested that the middle Eocene part makes up the bulk of the Eocene succession (e.g. Glaus, 1965), and volcanism climaxes during middle Eocene time (Berberian and King, 1981; Davoudzadeh et al., 1997; Verdel, 2009)."* For Middle Miocene see comment above. Concerning the other studies, if we exclude Verdel, what kind of data did the use? Paleontology? I think that this should be specified. If there is not any clear data-based evidence (like paleontology, possibly revised according to more recent biostratigraphy schemes), it will sound like a personal opinion, that probably fits with the general idea, but without data it will be just an opinion. In that case, I would probably remove that sentences.

- The authors write in the main text:
*Indeed, the Eocene volcanism in Iran erupted in shallow marine basins, and through significant amounts of carbonate-rich rocks of Jurassic, Cretaceous, and Paleogene age (e.g. Berberian and King, 1981).*
I think that the point is whare are the magmatic chambers and how long does it take for the magma to get to the surface. Yes, there are limestones in the stratigraphic sequence (it is the case in most of the sttings, if not all of the settings), but if the magma move quickly through these carbonaceous strata, I would not expect that much interaction. You may look at cases were a clear link between magmatism and interaction with the host rock was demonstrated. What conditions did you have? Are these conditions respected also in Iran?

- Finally, I must confess that I did not like the answer to the following comment
"Curiously, if the authors look at the Zachos et al., curve (δ18O curve vs age), they will see that the flare up in Iran coincides with the progressive Eocene cooling that culminates with the sharp temperature drop at the Eocene-Oligocene boundary (actually I think that such a curve, which is the base of all paleoclimatic reconstructions, should be shown also in this manuscript). To me this lack of correlation suggests that, although voluminous, the entire magmatic flare up in Iran did not have a strong impact on global climate, or at least that did not produce a change in the long-term global cooling trend. "
The comment intended to say that despite a prolonged phase of Eocene magmatism (ca. 55 or 50 Ma to ca. 38 Ma), we had a trend toward a colder climate (see Zachos curve), so it seems that there is a weak correlation  between climate and magmatism in Iran (unless we demonstrate that all magmatism occurred around the MECO). I am aware the authors did their best to use available data to demonstrate that magmatism may have peaked around 42-40 and I am fine with their efforts. However, they replied

*"As our radiometric age compilation shows in Figure 2C, the amount of radiometric ages drops rapidly between 35 and 32 Ma (the slope is nearly flat here). Also Figure 3 shows a drastic drop in igneous activity*

*around the Eocene-Oligocene transition. The flare-up in Iran precedes the Eocene-Oligocene transition by millions of years. On a side note, several of the authors are marine stratigraphers and paleoceanographers with ample experience on both the MECO and the Eocene-Oligocene Transition."*

Now, what is the link between a trend that seems to be in contrast with the flare up and the excellent experience of the authors in the field of MECO and the Eocene-Oligocene transition (note that the Eocene-Oligocene transition was just mentioned, but was not the focus of the comment)? I personally find this answer quite arrogant and irritating.

Anyway, I am happy to see that my comments helped.
Best regards.

---

## Author Response (AR2)

**Reviewer #1**

Dear van der Boon,

Thanks for the clarification. I agree with most of your points. Here I have two more suggestions that may help you to strengthen your points.

1. Since you have collected many age data with GPS information and highlighted your new data in figure 1b, I suggest that you plot all available mid-Eocene age on your figure 1a. The time-space distribution of the mid-Eocene rock in Iran would help readers to understand how widespread the Eocene rocks.

*This is a good suggestion and we have attempted to do this. However, we found that the more than 420 radiometric ages from 72 studies clutter up the figure so much that it becomes unreadable, and/or font too small to read. To accommodate for the suggestion of the reviewer, we have added a file to the supplementary information that contains all the ages of the supplementary information with their GPS locations. This file is our new supplementary file S4, and we have added to the text in section 7 "S4 is a .kmz file that contains the GPS locations of the literature ages (except of Shafaii Moghadam et al., (2020), who did not provide GPS locations), and can be opened in Google Earth."*

2. New age clusters at 40 Ma is in indeed close to the MECO. However, the age cluster does not guarantee the large volume of the volcanic eruption. If a net input of $CO_2$ causes the MECO, you would expect increasing volcanism from Paleocene (or early Eocene) to 40 Ma. Since you have the shapefile information, I suggest you make a plot: Area of volcanism vs. age. If the volume of the middle Eocene gets to the maximum, which would make your argument more convincing.

*We do not fully understand this comment. A pulse increase in volcanism at ~40 Ma is required to explain the MECO warming. In addition, it seems to us that the plot suggested by the reviewer is our Figure 2C.*

**Reviewer #3**

The manuscript "Exploring a link between the Middle Eocene Climatic Optimum and Neotethys continental arc flare-up" suggests the existence of a possible causal relationship between the Eocene arc volcanism associated with the subduction of the Neotethys ocean and the Middle Eocene Climatic Optimum (MECO). To test this hypothesis the authors provides new Ar-Ar ages of Middle Eocene volcanics exposed in NW Iran, compiles available Ar-Ar and Zircon U-Pb ages of Cenozoic intrusive and effusive rocks exposed all around Iran, and estimate the volume of $CO_2$ that might have been released in the atmosphere during the flare up. This working hypothesis represents an intriguing idea, however, it has not been clearly demonstrated in this version of the manuscript.

My main point is the same raised by Reviewer 1 "the successful link between the arc volcanism and climate change depends on how much carbon dioxide has been outputted" around the MECO. Furthermore, Reviewer 1 wrote: "Clearly, the authors have much overestimated the thickness of 40 Ma volcanic rocks. According to figure 2, we see volcanic events throughout the whole Eocene. Although there is an intense event at 40 Ma, still, the 40 Ma volcanic rocks are only a part of the Paleogene volcanic strata (3-9 km). You must be precise how thick is the 40 Ma rock." This is a key point that the authors did not address.

*Our main reply to this reviewer's comment is that we cannot prove, nor do we intend to, that the volcanism in Iran **caused** the MECO. We have phrased carefully to convey that point, both in the title*

*and throughout the manuscript. We explore whether this volcanism could have had an impact on global climate in the middle Eocene, since there has been a decade long search for volcanism that coincided with the MECO. We are open and honest about the limitations of this study, and present those clearly in the manuscript. Although the constraints that currently exist on both the timing and the areal extent of Eocene volcanism in Iran prevent a definitive conclusion on a causal relationship, they are good enough to make the 'back of the envelope' calculations that we present in this manuscript, but, again, the uncertainties are large and, in our view, clearly described. In the very least, we feel that the data compilation that we present here is large enough to support our assessment of the Eocene flare-up in Iran to broadly correspond to the MECO.*

The authors replied that: "Berberian & King (1981) state that "Extensive volcanism, with a wide range of composition, started in the Eocene Period (50 Ma) and continued for the rest of the period with the climax in Middle Eocene time (about 47-42 Ma). Despite their great thickness (locally up to 6 and 12 km) and wide distribution, the volcanics and tuffs were formed within a relatively short time interval." I have not found in the original paper any analytical data supporting this conclusion. Furthermore the authors replied that: "In the Alborz and Central Iran, middle Eocene extrusive volcanic formations are reported to be very thick, with estimates ranging from 3-5 km in the Alborz Mountains (Stöcklin, 1974), to 6-12 km locally throughout nearly all of Iran (Berberian and King, 1981). More recent estimates of the thickness are 3-9 kilometers (e.g. Morley et al., 2009; Verdel et al., 2011" This new text does not indicate the thickness of volcanics and volcaniclastics deposited around the MECO (or let's say at 40±2 Ma).

*Unfortunately, the amount of radiometric ages obtained on volcanic rocks in Iran is currently not large enough to assess exactly which part of the volcanic succession erupted within 2 million years around 40 Ma. This remains one of the uncertainties, as described in section 5. With regards to the thickness of the Eocene volcanic rocks, much of the mapping has been done in the 1960's, and many of the reports are not available online, or even in English. However, there are more assessments of thickness based on geological maps and cross-sections, and we have added a reference to Iwao and Hushmand-Zadeh (1971), who show a generalised lithostratigraphic column of the Karaj Formation in the Alborz Mountains. This column shows that the Karaj Fm has a thickness of around 10 kilometres. We have added to the text the following paragraph and references (lines 149-158): "These estimates are supported by geologic maps and their descriptions that are based on extensive fieldwork. Estimates from maps range mostly between 2 and 7 kilometres. On the lower side are Saein Qaleh (Kholghi Khasraghi, 1994), Saveh (Ghalamghash et al., 1998b), Kuhpayeh (Radfar et al., 2002), with thicknesses of ~2 kilometres, Tafresh (Hadjian et al., 1999) with ~3 kilometres, then Meyamey (Amini Chehragh and Ghalamghash, n.d.), Tarom (Hirayama et al., 1966) and Kalateh (Jafarian, n.d.) with around ~4 kilometres, while Kajan (Amini and Amini Chehragh, 2001), Kahak (Ghalamghash et al., 1998a) and Lahrud (Babakhani et al., 1991) mention thicknesses of the Eocene volcanic succession of around ~6 kilometres, and Bardsir (Mohajjel Kafshdouz and Khodabandeh, 1992) of around 7 kilometres. On the other hand, Iwao and Hushmand-Zadeh, (1971) show a generalised lithostratigraphic log of the Karaj formation, and mention that the succession reaches a thickness of more than 10 kilometres in the Alborz Mountains." We have adjusted our Table 2 to include more estimates of thicknesses, and we calculate $CO_2$ estimates for the entire range of thicknesses between 2 and 10 kilometres. We have added to the text (lines 158-160) "In Table 2, we calculate how much $CO_2$ could have been released through formation of different volumes of volcanic rocks. We calculate this for a range of thicknesses between 2 and 10 kilometres." and (lines 188-192) "Table 2 shows that middle Eocene volcanic rocks with thicknesses between 2 and 7 kilometres, and a contribution from limestones, give estimates that lie within the range expected for the MECO (marked in green in Table 2). There could have been a contribution to $CO_2$ through skarn formation by intrusive activity, which clusters around 40.5 Ma (see Figure 3D), although we currently lack the constraints to quantitatively assess this." and (lines 188-190): "Table 2 shows that middle Eocene*

*volanic rocks with thicknesses between 2 and 7 kilometres, and a contribution from limestones, give estimates that lie within the range expected for the MECO (marked in green in Table 2).".*

In section 4 the authors calculate the volume of volcanic rocks that have been erupted during the flare up (3-9 km of thickness times 40.000 km2) and based on a "linear relationship of a lava volume" (see line 150) they estimate the amount of released CO2. It is not clear to me what is the age of the Middle age volcanics considered in the calculation. From figure 2 and from the text I have the impression that the authors considered all the magmatic rocks produced during the entire flare up, which lasted about 20 million years (from ca. 55 to 35 Ma), and not the thickness of the 40±2-My-old magmatic rocks. In the text, the authors recall also table 2, but I could not find it in the text. As specified above, the amount of magmatic rocks emplaced around the MECO is a crucial point. Without an idea of such a thickness, the released CO2 cannot be estimated, and the working hypothesis cannot be tested.

*We explained in lines 118-122 that we calculate the amount of middle Eocene volcanic rocks in the following way: "Estimation of the areal extent of middle Eocene volcanic rocks is done using the shapefiles of Sahandi et al. (2014). For the Eocene, shapefiles are classified as 'Eocene', 'Eocene-Oligocene', 'Late Eocene-Oligocene', 'Middle Eocene', and 'Middle-Late Eocene'. We assumed that shapefiles specified as 'Eocene' had the same proportion of middle Eocene igneous rocks, and thus calculated an areal extent of 38223 $km^2$ of middle Eocene igneous rocks." So we calculate estimated volumes for the middle Eocene (10 Myr), as this is the maximum resolution that we can get from the shapefiles. With only one radiometric age for every few hundred square kilometres in Iran, this is currently as good as it will get without a stellar amount of new radiometric ages. We have now rephrased this, in the hope that it is clearer. It now reads (lines124-130): "According to the geologic maps, 54% of all area covered by volcanic rocks in Iran is of Eocene age. For the Eocene, shapefiles are classified as 'Eocene', 'Eocene-Oligocene', 'Late Eocene-Oligocene', 'Middle Eocene', and 'Middle-Late Eocene'. More than half is marked as 'Eocene' and not specified further, but of the rest that is specified, almost half is 'Middle Eocene'. Assuming that the unspecified Eocene rocks have approximately the same age distribution as the specified Eocene rocks, we estimate that roughly half of the Eocene volcanic rocks in Iran and a quarter of the total area covered by volcanic rocks in Iran is of middle Eocene age. We use these areas to estimate the volumes of volcanic rocks formed in the middle Eocene."*

Curiously, if the authors look at the Zachos et al., curve (δ18O curve vs age), they will see that the flare up in Iran coincides with the progressive Eocene cooling that culminates with the sharp temperature drop at the Eocene-Oligocene boundary (actually I think that such a curve, which is the base of all paleoclimatic reconstructions, should be shown also in this manuscript). To me this lack of correlation suggests that, although voluminous, the entire magmatic flare up in Iran did not have a strong impact on global climate, or at least that did not produce a change in the long-term global cooling trend.

*As our radiometric age compilation shows in Figure 2C, the amount of radiometric ages drops rapidly between 35 and 32 Ma (the slope is nearly flat here). Also Figure 3 shows a drastic drop in igneous activity around the Eocene-Oligocene transition. The flare-up in Iran precedes the Eocene-Oligocene transition by millions of years. On a side note, several of the authors are marine stratigraphers and paleoceanographers with ample experience on both the MECO and the Eocene-Oligocene Transition.*

If the authors want to demonstrate a causal relationship between arc volcanism and MECO they need to document an increase in magmatic flux at 40±2 Ma. So far, nobody as really demonstrated it. To me, at least as first approach, they should look at few stratigraphic sections all around the country, measure their thickness, extract the depositional ages and then estimate changes in

magmatic flux through time. I think that is the most direct way to test such a working hypothesis. After that, they may look at the areal distribution on Middle Eocene rocks, assuming that these are really Middle Eocene rocks that were deposited around the MECO and not during the entire flare up.

*We fully agree. We have incorporated this point in section 5 (lines 214-215): "Acquisition of radiometric ages throughout sections that cover the entire Eocene volcanic succession could aid in quantification of magmatic flux over time."*

Indeed, the Peshtasar Formation is a good target because available ages from Vincent et al (2005), recalibrated by the authors, indicate deposition of an up to 1.4-km-thick sequence of volcanics and volcaniclastics between 41 and 39 Ma. Note that we are talking about 1.4 km of volcanics and volcaniclastics and not 3 to 9 km. Similar work should be done in other areas.

*The Peshtasar formation is certainly important but, importantly, it is only **part of** the entire (middle?) Eocene succession in the Talysh Mountains. The Peshtasar Formation is more than 2 kilometres thick (van der Boon et al., 2017) and consists only of basalts and sills, likely formed in three very short pulses of intense volcanism. There is another formation underneath this, the Kosmalyan formation, which also consists of another large amount of volcanic and volcaniclastic rocks. Although this formation has not been dated, nor studied in detail, Vincent et al. (2005) estimate the thickness of the Kosmalyan formation as more than 7 kilometres, with an estimated age of late early to early middle Eocene. This could thus mean that the middle Eocene succession in the Talysh of Azerbaijan is around 9 kilometres thick, which would be in line with findings from the Talesh and Alborz of Iran. Collectively, we passionately agree with the reviewer that the entire Eocene succession in Iran warrants detailed study and hope that our paper will spur enthusiasm of the geology community to study Eocene volcanic rocks in Iran.*

I understand that the compilation provided by the authors try to overcome the paucity of stratigraphic information available in literature, but that strategy is biased toward the sampled stratigraphic intervals. It may be correct for intrusive rocks because there might be a cluster of ages (assuming that the cooling recorded by the Ar-Ar system occurred within 1-2 Million years) indicating a specific episode of magma emplacement and possibly an overproduction of magma. However, according to available data (Verdel et al., 2011, is probably the best reference) effusive and pyroclastic rocks could have been deposited between 55 and 35 Ma at rather uniform rates . Except the work of Vincent et al., 2005, there are no studies pointing toward an increase in the magmatic flux around the MECO.

*We agree with the reviewer that our compilation is likely biased towards certain accessible areas and intervals, and we have added to the text the following paragraph (lines 196-200): "Despite the fact that sampling biases (i.e. sampling is often focused on easily accessible sites and certain time periods) can never be avoided, our compilation of radiometric ages shows a good correlation to the geologic maps, in the sense that the radiometric ages confirm that the flare-up took place during the middle Eocene. We note that the Miocene peak (Figure 3C) is relatively high compared to the Eocene, which could be caused by a sampling bias, as the geologic maps (Sahandi et al., 2014) indicate that only 2-4% of Iran is covered by Miocene volcanic rocks." We disagree with the reviewer that there are no studies besides Vincent et al. (2005) that point towards an increase in magmatic flux around the MECO. On the contrary, field studies have often suggested that the middle Eocene part makes up the bulk of the Eocene succession (e.g. Glaus, 1965). Also Davoudzadeh et al. (1997) mention "Extensive volcanism with a wide range of composition started in Upper Cretaceous and continued throughout the Cainozoic with the climax in Middle Eocene time." In our previous response to reviewer 1, we also mentioned that Berberian & King (1981) state that "Extensive volcanism, with a wide range of composition, started in the Eocene Period (50 Ma) and continued for the rest of the period with the*

*climax in Middle Eocene time (about 47-42 Ma). Despite their great thickness (locally up to 6 and 12 km) and wide distribution, the volcanics and tuffs were formed within a relatively short time interval." Furthermore, Verdel (2009) shows in Figure 5 of Chapter 3 that most of the flare-up is during the middle Eocene for North, West and East Iran, and only in Central Iran extends also into the lower Eocene. Additionally, a middle Eocene increase in volcanism can be seen from the slope of the radiometric ages in our plot in Figure 2C, which is much steeper during the Lutetian and Bartonian, indicating that there are many more radiometric ages in this time period than in the times before and after the middle Eocene. We have added to the introduction (lines 58-60): "Field studies have often suggested that the middle Eocene part makes up the bulk of the Eocene succession (e.g. Glaus, 1965), and volcanism climaxes during middle Eocene time (Berberian and King, 1981; Davoudzadeh et al., 1997; Verdel, 2009)."*

Finally, concerning the compilation and the genesis of magma (see comment 1 of Reviewer 2), I suggest looking at a recent publication of Rabiee et al., in Gondwana Research titled: "Long-lived, Eocene-Miocene stationary magmatism in NW Iran along a transform plate boundary"; there, new ages of intrusives and a similar compilation is provided.

*We thank the reviewer for this suggestion. Consequently, we have added the U-Pb ages of this and of another 11 studies (Almasi et al. (2019), Etemadi et al. (2020), Javidi Moghadam et al. (2019, 2020), Khaksar et al. (2020), Maleki et al. (2019), Mazhari et al. (2020), Rabiee et al. (2019, 2020), Sepidbar et al. (2019), Shafaii Moghadam et al. (2020) and Simmonds (2019)) to our compilation, which now consists of 72 papers and more than 420 ages. We updated our Supplementary files S2, S3 and S33 accordingly. We made new versions of figures 2C and figures 3B, 3C and 3D. We have adapted ages for the peaks in the text accordingly.*

Here are few minor points:
1) In the abstract, the authors suggest that magma emplacement in carbonaceous rocks may have increased the total amount of $CO_2$ released. This is not really addressed in the text except in lines 159-161 where the authors write: "Indeed, the Eocene extrusive volcanism in Iran erupted through significant amounts of carbonate-rich rocks of Jurassic, Cretaceous, Paleogene age (e.g. Berberian and King, 1981)". By looking at geologic maps in NW Iran, it seems to me that most of Eocene intrusions are found within Eocene volcanics and volcaniclastics (meaning that they intruded at shallow depth) rather than in Paleo-Mesozoic carbonates (while Paleogene carbonates are rather thin). I cannot see evidences of intrusions in carbonaceous lithology based on available geologic maps. It may be true, with erosion that has not brought yet these rocks at the surface, but currently there is not any evidence for that.

*We do not really understand this comment. The reviewer refers to our lines 159-161, where we talk about extrusive rocks, but then continues to state that Eocene intrusions are found within Eocene volcanics and volcaniclastics. We agree with the reviewer and there is no contrast here. We have however removed the pleonasm 'extrusive volcanism' throughout the manuscript.*

*We have, however, added some more references and a paragraph on the association of the volcanic rocks with shallow marine carbonate-rich rocks, which now reads (lines 179-184): "Indeed, the Eocene volcanism in Iran erupted in shallow marine basins, and through significant amounts of carbonate-rich rocks of Jurassic, Cretaceous, and Paleogene age (e.g. Berberian and King, 1981). Glaus, (1965) mentions that middle Eocene limestones occur as lenticular masses within the basaltic flows, or as consistent horizons associated with tuffs. Verdel (2009) shows that Eocene volcanic rocks are formed in close association with Eocene limestones in north, west and east Iran. This is also the case in central Iran, which can be seen from geologic maps, such as the one from Qom (Emami, 1981)."*

2) Reviewer 1 suggested also to look also at other regions as possible sources of arc volcanism around the MECO. Of course, the first region to look at would be the entire Middle East, which represents the upper plate of the Neothetys subduction system (Turkey, Armenia, Georgia, Azerbaijan). The authors replied "Unfortunately, the lack of shapefiles of Eocene volcanic and intrusive rocks in Armenia and Azerbaijan, along the Lesser Caucasus Mountains (e.g. Allen and Armstrong, 2008), and plutons and volcanic rocks in Armenia (e.g. Moritz et al., 2016; Sahakyan et al., 2016), hampers calculations on additional CO2 emissions within these regions"
I do not think that the lack of shape files hampers the calculations. The lack of a geologic map with clear ages of volcanics hampers the calculations. If you have a geologic map with good age information you can easily digitize the contours of the Middle Eocene volcanics and create a shape file.

*These are all good suggestions and this would be good follow-up study. Unfortunately, we do not have such maps. The maps we have for Azerbaijan and Armenia are very low resolution (and we only have one of each country), and stand in stark contrast to the more than 500 maps we have from Iran.*

Note, however, that there is still a large volume of volcanic and volcaniclastic rocks buried below late Cenozoic sediments that are difficult to estimate. This is particularly true in Central Iran where depositional processes are dominant and localized exhumation hasn't yet exposed the Eocene volcanics. This means that any estimates based on outcrops will be always a very minimum value.

*We agree with the reviewer, and mention this in lines 174-178 (new lines 192-196): "Erosion has affected the entire Iranian plateau, and could have eroded away significant volumes of Eocene volcanic rocks. Morley et al. (2009) and Ballato et al. (2011) note that clasts in the Lower and Upper Red formation (Oligocene-Miocene age), which in many places overlie Eocene volcanics, are for a large part made up of eroded Eocene volcanic rocks. Original thicknesses of Eocene volcanic rocks in Iran could thus have been larger, making our $CO_2$ output estimate a minimum estimate".*

Finally, I am not a geochemist, so I cannot comment on the calculations for estimating the released CO2, but I guess that the starting point is a reliable estimate of the volume of volcanics and volcaniclastics ejected around the MECO.

I hope these comments will help. Good luck.

*We thank the reviewer for their comments that have prompted us to rethink a number of issues, further complete our database and improve our manuscript.*

[revised manuscript text omitted]

---

## Author Response (AR3)

I have read through the updated version of "Exploring a link between the Middle Eocene Climatic Optimum and Neotethys continental arc flare-up" by Van der Boon et al., and the response to my original comments. I recognize that the authors have made a good faith effort to address my original concerns and that some assumptions are now better motivated (especially on the thickness of the Eocene rocks). I still disagree with some answers and observations. Please see below:

- The authors write as reply to my comment:
*"Furthermore, Verdel (2009) shows in Figure 5 of Chapter 3 that most of the flare-up is during the middle Eocene for North, West and East Iran, and only in Central Iran extends also into the lower Eocene."* Figure of Verdel's thesis shows 3 ages for Central Iran 52.9±3.3, 50.0±4.4, 54.7±3.1, and one age for northern Iran 52.2±3.4 Ma; all these ages fall within the error, so it is not clear to me what is the age of the Karaj Formation (maybe we should look at paleontological works on the Ziarat Formation). In any case the main point is to have an increase in magmatic flux around 42-40 Ma. Intense magmatism between 50 and 45 Ma or 55 and 45 Ma doesn't help. To have an impact on MECO should be coeval and I guess that ,despite all difficulties, your data seems to go in that direction.

We are happy that the reviewer agrees that our data show an increase in magmatic flux around MECO times. We realize and agree that there is also evidence for older volcanism in Iran, but we clearly show here that it peaks around 40 Ma.

- The authors write in the main text:
*"Field studies have often suggested that the middle Eocene part makes up the bulk of the Eocene succession (e.g. Glaus, 1965), and volcanism climaxes during middle Eocene time (Berberian and King, 1981; Davoudzadeh et al., 1997; Verdel, 2009)."* For Middle Miocene see comment above. Concerning the other studies, if we exclude Verdel, what kind of data did the use? Paleontology? I think that this should be specified. If there is not any clear data-based evidence (like paleontology, possibly revised according to more recent biostratigraphy schemes), it will sound like a personal opinion, that probably fits with the general idea, but without data it will be just an opinion. In that case, I would probably remove that sentences.

We assume that the reviewer means the middle Eocene here and not the Middle Miocene.

Indeed, this is based mostly on biostratigraphic data. For example the presence of *Nummulites perforatus* in the Karaj Fm (e.g. Sieber, 1970) indicates a middle Eocene age (Laura Cotton, personal communication). We agree with the reviewer that this could be better dated. Nonetheless, our data compilation clearly indicates that the peak of volcanism happens during the middle Eocene. We have adjusted the sentence so it now reads: "*In the past, it has been hypothesised based on field studies, that the middle Eocene part makes up the bulk of the Eocene volcanic succession (e.g. Glaus, 1965), because of the presence of middle Eocene nummulites within the Karaj formation (e.g. Sieber, 1970) and volcanism climaxes during middle Eocene time (Berberian and King, 1981; Davoudzadeh et al., 1997; Verdel, 2009).*"

- The authors write in the main text:
*Indeed, the Eocene volcanism in Iran erupted in shallow marine basins, and through significant amounts of carbonate-rich rocks of Jurassic, Cretaceous, and Paleogene age (e.g. Berberian and King, 1981).*
I think that the point is whare are the magmatic chambers and how long does it take for the magma to get to the surface. Yes, there are limestones in the stratigraphic sequence (it is the case in most of the sttings, if not all of the settings), but if the magma move quickly through these carbonaceous strata, I would not expect that much interaction. You may look at cases were a clear link between

magmatism and interaction with the host rock was demonstrated. What conditions did you have? Are these conditions respected also in Iran?

The reviewer is correct here. However, the lack of geological constraints makes it currently impossible to quantitatively address this question. Future detailed studies could massively aid in assessing these conditions, and we hope that our hypothesis of a possible "link between the Middle Eocene Climatic Optimum and Neotethys continental arc flare-up" will encourage people to undertake this kind of research. We have added to section 5: "*Studies that focus on the interaction between carbonates and magma chambers could aid in quantifying the carbonate contribution to $CO_2$ release.*"

- Finally, I must confess that I did not like the answer to the following comment
"Curiously, if the authors look at the Zachos et al., curve (δ18O curve vs age), they will see that the flare up in Iran coincides with the progressive Eocene cooling that culminates with the sharp temperature drop at the Eocene-Oligocene boundary (actually I think that such a curve, which is the base of all paleoclimatic reconstructions, should be shown also in this manuscript). To me this lack of correlation suggests that, although voluminous, the entire magmatic flare up in Iran did not have a strong impact on global climate, or at least that did not produce a change in the long-term global cooling trend. "
The comment intended to say that despite a prolonged phase of Eocene magmatism (ca. 55 or 50 Ma to ca. 38 Ma), we had a trend toward a colder climate (see Zachos curve), so it seems that there is a weak correlation between climate and magmatism in Iran (unless we demonstrate that all magmatism occurred around the MECO). I am aware the authors did their best to use available data to demonstrate that magmatism may have peaked around 42-40 and I am fine with their efforts. However, they replied
*"As our radiometric age compilation shows in Figure 2C, the amount of radiometric ages drops rapidly between 35 and 32 Ma (the slope is nearly flat here). Also Figure 3 shows a drastic drop in igneous activity around the Eocene-Oligocene transition. The flare-up in Iran precedes the Eocene-Oligocene transition by millions of years. On a side note, several of the authors are marine stratigraphers and paleoceanographers with ample experience on both the MECO and the Eocene-Oligocene Transition."*
Now, what is the link between a trend that seems to be in contrast with the flare up and the excellent experience of the authors in the field of MECO and the Eocene-Oligocene transition (note that the Eocene-Oligocene transition was just mentioned, but was not the focus of the comment)? I personally find this answer quite arrogant and irritating.

We did not intend to upset the reviewer and apologise for the misunderstanding. It is beyond the goal of our study to explain the entire Zachos curve for the Eocene, and we agree with the reviewer that the flare-up is likely longer than the duration of the MECO, and we discuss the obstacles to linking volcanism to the MECO in section 5. The drastic cooling around the Eocene-Oligocene transition happens after peak volcanism has ended, as our data show. We speculate there could be a link between weathering of the huge amounts of volcanic rocks in Iran and global cooling, but this requires additional research and is beyond the scope of this paper.

Anyway, I am happy to see that my comments helped.
Best regards.

We thank the reviewer again for their comments that have further improved our manuscript.

[revised manuscript text omitted]